# Abdominal organ injury in cardiac arrest: Systematic literature review

**Bjørn Hoftun Farbu** [1,2,3]*, **Jostein Hagemo** [3,4,5], **Marius Rehn** [3,4,5]

1 Department of Anaesthesiology and Intensive Care Medicine, St. Olav's University Hospital, Trondheim, Norway, 2 Institute of Circulation and Medical Imaging, Faculty of Medicine and Health Sciences, Norwegian University of Science and Technology (NTNU), Trondheim, Norway, 3 Department of Research and Development, Norwegian Air Ambulance Foundation, Oslo, Norway, 4 Division of Prehospital Services, Oslo University Hospital, Oslo, Norway, 5 Institute of Clinical Medicine, University of Oslo, Oslo, Norway

* bjorn.farbu@gmail.com

## Abstract

### Background

Both cardiopulmonary resuscitation (CPR) and ischaemia could lead to abdominal organ injury. However, the importance of abdominal injury in cardiac arrest remains uncertain. We aimed to systematically review indexed literature to describe incidence of abdominal injury after non-traumatic cardiac arrest and associations with outcome.

### Methods

We searched MEDLINE/PubMed, Embase, The Cochrane Database of Systematic Reviews and Scopus up to 12th September 2024 for studies reporting differences in outcomes between patients with and without abdominal injury, and all studies reporting abdominal adverse events after cardiac arrest. Two independent reviewers screened articles for eligibility. One reviewer extracted data and assessed risk of bias using the Critical Appraisal Skills Programme checklist. Injuries were defined as traumatic or ischaemic, either in the studies or otherwise by the reviewers. Results were summarized and presented in tables and Forest plots. We followed the PRISMA guidelines, and registered the study in PROSPERO.

### Results

We included 68 studies and 140 case reports. Most studies were single-centre. Quantitative synthesis of evidence was not feasible given high heterogeneity and risk of bias. Traumatic injuries affected mostly liver and spleen, with incidences from 0% to 15%, reaching 29% in one study of mechanical chest compressions. Life-threatening injuries were uncommon. The incidence of ischaemic injury was dependent on assessment method; 7% to 28% had liver injury, 0.7% to 2.5% was

**Data availability statement:** All relevant data are within the manuscript and its Supporting information files.

**Funding:** The Norwegian Air Ambulance Foundation funded the review. The funders had no role in study design, data collection and analysis, decision to publish, or preparation of the manuscript.

**Competing interests:** The authors have declared that no competing interests exist.

**Abbreviations:** CT, Computed tomography; CPR, Cardiopulmonary resuscitation; ECPR, Extracorporeal cardio-pulmonary resuscitation; FAST, Focused Assessment With Sonography in Trauma; ICU, Intensive care unit; IFABP, Intestinal fatty acid binding protein; IHCA, In-hospital cardiac arrest; LUCAS, Lund University Cardiopulmonary Assist System; MeSH, Medical Subject Headings; NOMI, Non-occlusive mesenteric ischaemia; OHCA, Out-of-hospital cardiac arrest; PICOS, Population, Intervention/Exposure, Comparator, Outcome, Study design; PRISMA, Preferred Reporting Items for Systematic Reviews and Meta-analyses; RCT, Randomised controlled trial; REBOA, Resuscitative endovascular balloon occlusion of the aorta; ROSC, Return of spontaneous circulation.

diagnosed with non-occlusive mesenteric ischaemia, 82% to 100% had intestinal injury measured by biomarkers. Ischaemic injuries were associated with mortality.

## Conclusion

In this comprehensive review of abdominal injuries following cardiac arrest, CPR-related traumatic injuries were uncommon, but should be considered in patients with unexplained clinical deterioration. Ischaemic injury incidence ranged from 0.7% to 100%, and was consistently associated with mortality. Whether abdominal ischaemia independently contributes to poor outcomes remains unresolved and warrants further investigation.

PROSPERO ID: CRD42022311508.

## Introduction

Patients who are successfully resuscitated from cardiac arrest frequently develop multiple organ failure. In the initial days following resuscitation, circulatory failure is the leading cause of death, while hypoxic-ischaemic brain injury accounts for the majority of deaths overall. The pathophysiologic mechanisms linking cardiac arrest to multiple organ failure are incompletely understood, but are believed to involve a systemic ischaemia-reperfusion injury [1].

The role of abdominal injury after cardiac arrest remains poorly defined. Liver injury has been associated with circulatory failure and poor neurological outcome, but has not been consistently linked to mortality [2,3]. Ischaemic lesions of the upper gastro-intestinal tract has been reported as common and may correlate with unfavorable neurolocial outcome [4]. Additional studies have demonstrated associations between intestinal injury and increased risk of organ dysfunction, poor neurological outcome, and death [5–7]. Administration of adrenaline (epinephrine), the main drug of resuscitation, may exacerbate splanchnic ischemia via vasoconstriction of mesenteric vessels [4,6,8,9]. Moreover, cardiopulmonary resuscitation (CPR)-related traumatic injuries might be more common with mechanical chest compressions [10]. Both ischaemic and CPR-related abdominal injuries could contribute to ischaemia-reperfusion injury, circulatory failure and a poor outcome. Nonetheless, the incidence and prognostic implications of abdominal injury following non-traumatic cardiac arrest remain uncertain.

We aimed to systematically review indexed literature to characterize the incidence of abdominal injury after non-traumatic cardiac arrest and to assess its association with clinical outcomes.

## Materials and methods

### Literature search

We searched MEDLINE/PubMed, Embase, and The Cochrane Database of Systematic Reviews up to 12th September 2024, and conducted a citing reference search in Scopus up to 15th November 2024. There was no limitation for language or year of publication. Grey literature was not searched.

The study followed the Preferred Reporting Items for Systematic Reviews and Meta-analyses (PRISMA) guidelines including the PICO methodology (Population, Intervention/exposure, Comparator, Outcome) [11]. The PRISMA checklist is provided in S1 Appendix. The PICO question was: Do patients with adult or paediatric, out-of-hospital or in-hospital, non-traumatic cardiac arrest (P), with abdominal organ injury in any form (I), compared to patients without abdominal organ injury C), have different short- or long-term survival, neurological outcome at discharge and/or long-term, or other outcome as defined in the study (O). The protocol was registered in the international prospective register of systematic reviews, PROSPERO (ID CRD42022311508).

The search strategy consisted of both controlled terms (Medical Subject Headings [MeSH], Emtree) and free text words for two basic concepts: 1. Cardiac arrest and 2. Abdominal organ injury. The first set of entry terms described the patient group, and the second set described the exposure. We used Boolean operators for term combinations and included synonyms. Proximity operators and field specification were applied when available and suitable. The complete search strategy is available in S2 Appendix.

## Study selection

Records from the literature search were imported into the online Covidence tool [12]. Two pairs of authors (BHF paired with MR or JH) screened all titles and abstracts independently, and studies clearly not meeting the inclusion criteria were excluded. We reviewed potentially eligible studies in full text, and reasons for non-eligibility are reported in a PRISMA flow chart. Disagreements were solved through discussion until consensus was reached. Reference lists of included publications were searched manually.

Studies fulfilling one of the following two criteria were included: 1) Randomized trials, non-randomized controlled trials, observational studies (cohort studies and case-control studies) reporting data about differences in terms of clinical outcomes between patients with and without abdominal organ injury, or 2) Case reports/series and other studies when they reported abdominal adverse events. Exclusion criteria were animal studies, editorials, comments and letters to the editor, literature reviews, studies published solely as abstracts, and studies of mixed populations where exposure was not specifically stated for cardiac arrest patients.

We made the following amendments to protocol. "And other studies" was added to criterion 2) in the inclusion criteria, and we specified "abdominal" adverse events. We added three exclusion criteria (reviews, abstracts, mixed populations), but chose to not exclude articles without abstract. We present incidence rather than prevalence, and risk ratios rather than odds ratios. Quality appraisal was done by one reviewer.

## Data extraction and quality assessment

One reviewer (BHF) extracted data using the Covidence online tool with predefined variables regarding demographic, Utstein cardiac arrest data, exposure, treatment, and outcome. The same reviewer used the Critical Appraisal Skills Programme checklist for quality appraisal [13].

## Definitions

Cardiac arrest was defined as all situations where CPR had been performed, or otherwise as defined in the study. Traumatic cardiac arrest was defined as all cases caused by trauma, including electrocution and avalanche. Drowning and strangulation were deemed non-traumatic.

Abdominal organs in this study were liver, stomach, small and large intestine, rectum, pancreas, and spleen, but not kidney and urogenital organs. Any form of injury was part of the study, as judged by radiologic examination, surgery, endoscopy, biomarker, post-mortem examination, physicians discretion, or other injury as defined in the study.

Injuries were referred to as traumatic or ischaemic as defined in the studies. If not defined, the following injuries were referred to as traumatic: hollow viscus perforation, pneumoperitoneum, hemoperitoneum, tear, laceration, contusion, rupture, hematoma, and hemorrhage. Rise in biomarkers, e.g., transaminases, and mucosal lesions were referred to as ischaemic injury.

## Statistical methods

Incidences are presented as percentages when referring to the range among several studies, and percentage (proportion) when referring to separate studies. Risk ratio with 95% confidence intervals for mortality and organ injury according to the prespecified subgroup treated with mechanical compressions were calculated using 2x2 tables with exposure and outcome as dichotomised in the studies, and presented as Forest plots. If required data were unavailable, we contacted the corresponding authors. Heterogeneity was assessed through manual evaluation, without statistical estimation. Given the absence of a quantitative synthesis, only the data at hand were reported, and no formal steps were taken to address missing data. The statistical calculations were conducted with Stata 18.0 (StataCorp LCC, Collage Station, TX).

## Ethics

No ethics approval was indicated, as this was a literature review.

## Results

The study selection is presented in a PRISMA flow chart (Fig 1). The search retrieved 1783 studies, of which 328 were duplicates. Of 1460 publications screened for title and abstract, we assessed 329 studies in full-text for eligibility. Reasons for exclusion are presented in Fig 1. We included 208 studies in the review.

### Characteristics of included studies

Among the 208 studies, there were 52 retrospective and 10 prospective cohort studies, one case control study, one non-randomised and four randomised controlled trials (RCT) [2–7,14–75]. There were 140 case reports of abdominal injuries (references are provided in S3 Appendix). The cohort and controlled studies were all published in English, except one in Slovak, and came from the United States of America (n = 17), France (9), Germany (7), Austria (6), Republic of Korea (5), Japan (4), Czech Republic (2), Finland (2), Norway (2), Sweden (2), Switzerland (2), Turkey (2) and one study from Belgium, Canada, China, Denmark, the Netherlands, Republic of Singapore, Spain, and Thailand, respectively. Twenty-five studies were of out-of-hospital cardiac arrest (OHCA) patients, seven of in-hospital cardiac arrest (IHCA) patients, 21 studies of mixed OHCA and IHCA. Otherwise location of arrest remained unreported. The case reports came from all continents except Africa, and were written in a language familiar to the reviewers, or had an English abstract with sufficient information.

Due to a large number of studies in various populations, the studies were stratified post-hoc and are presented in the following six groups: Studies reporting abdominal injuries in general [14–34], studies comparing mechanical and manual chest compressions [35–46], organ specific studies [2–7,47–56], studies of extracorporeal cardiopulmonary resuscitation (ECPR) patients [57–64], paediatric studies [65–70], and miscellaneous [71–75]. Quantitative synthesis of evidence was not feasible given high heterogeneity among studies in participants, exposure, and outcome reported [76].

### Quality of included studies

Many studies were small, single-centre, and of various sub-populations, such as forensic studies or patients with registered abdominal computed tomography (CT). Most studies addressed a focused issue, but in several studies the measure of exposure was subjective and/or not clearly defined. Outcomes on the other hand were clearly defined. The results were often presented with wide confidence intervals. The quality appraisal is presented in S4 Appendix.

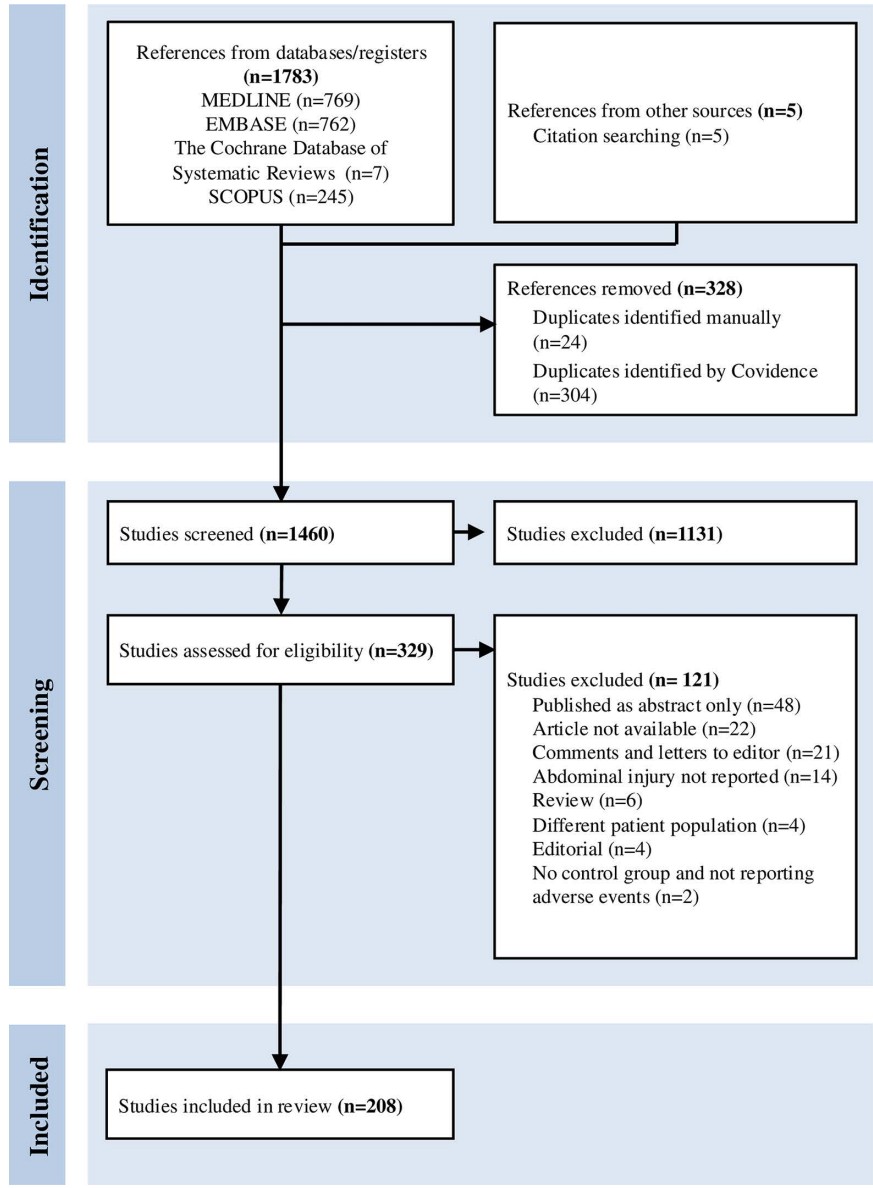

**Fig 1. PRISMA flow chart.**

## Incidence

CPR-related traumatic injuries were mostly diagnosed by CT or autopsy, and incidences of the most common injuries are presented in Fig 2. The risk ratios for abdominal injury in patients treated with mechanical compared to manual compressions, are presented as a Forest plot in Fig 3. Ischaemic injury were often diagnosed by biomarkers or endoscopy, but also CT and autopsy. The incidence was highly dependent on assessment method and are presented in Tables 1–3. Incidences of abdominal injury are also presented separately for each group in the main text.

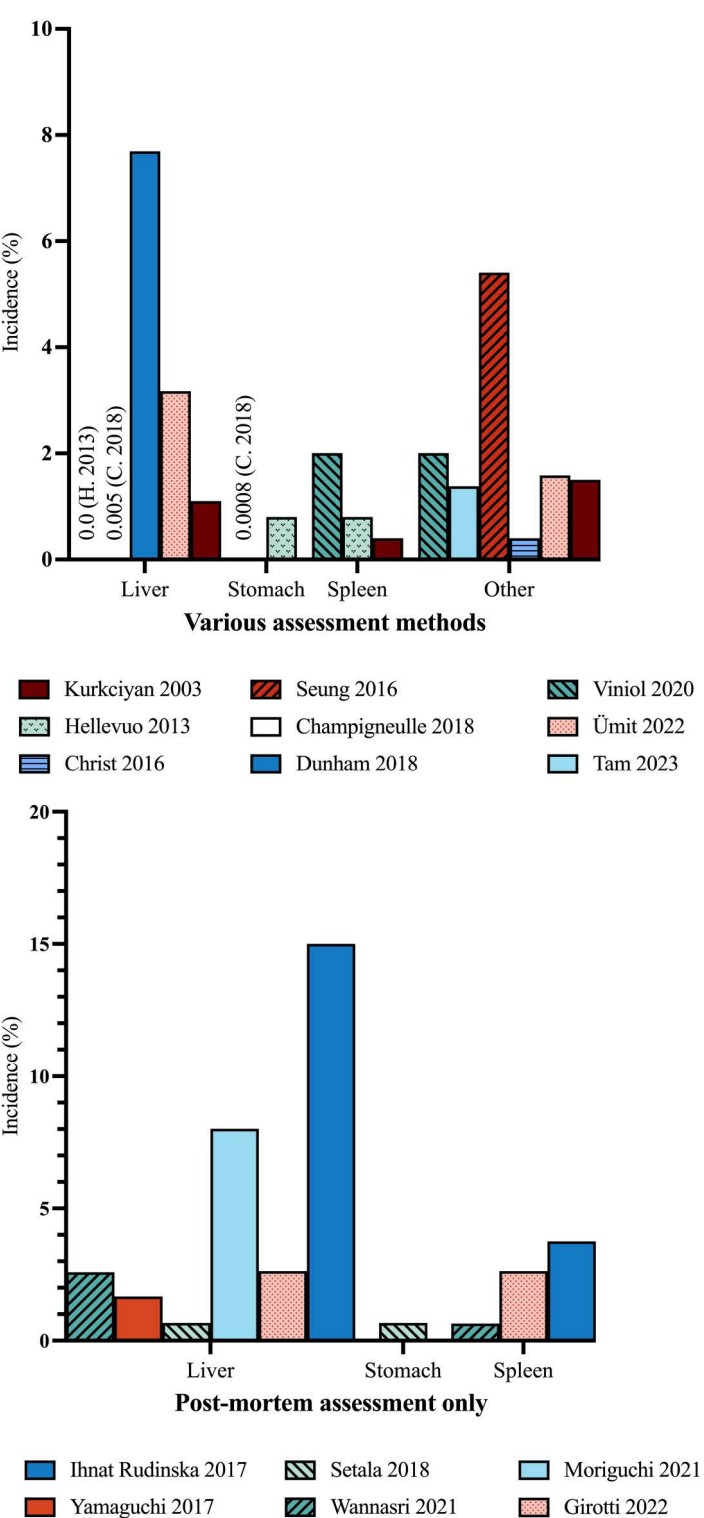

**Fig 2. Incidences of CPR-related traumatic abdominal organ injury.** Reported incidences of abdominal organ injury in general according to various assessment and post-mortem assessment only, respectively. Multimodal assessment includes one or more of the following: Medical records, computed tomography (CT), register data, and autopsy. Only incidences which are explicitly reported are shown. Injuries to organs not reported are not shown, either it was not looked for, or not found. "Other" injury: Duodenal perforation, pancreatic injury, intraperitoneal hemorrhage or pneumoperitoneum.

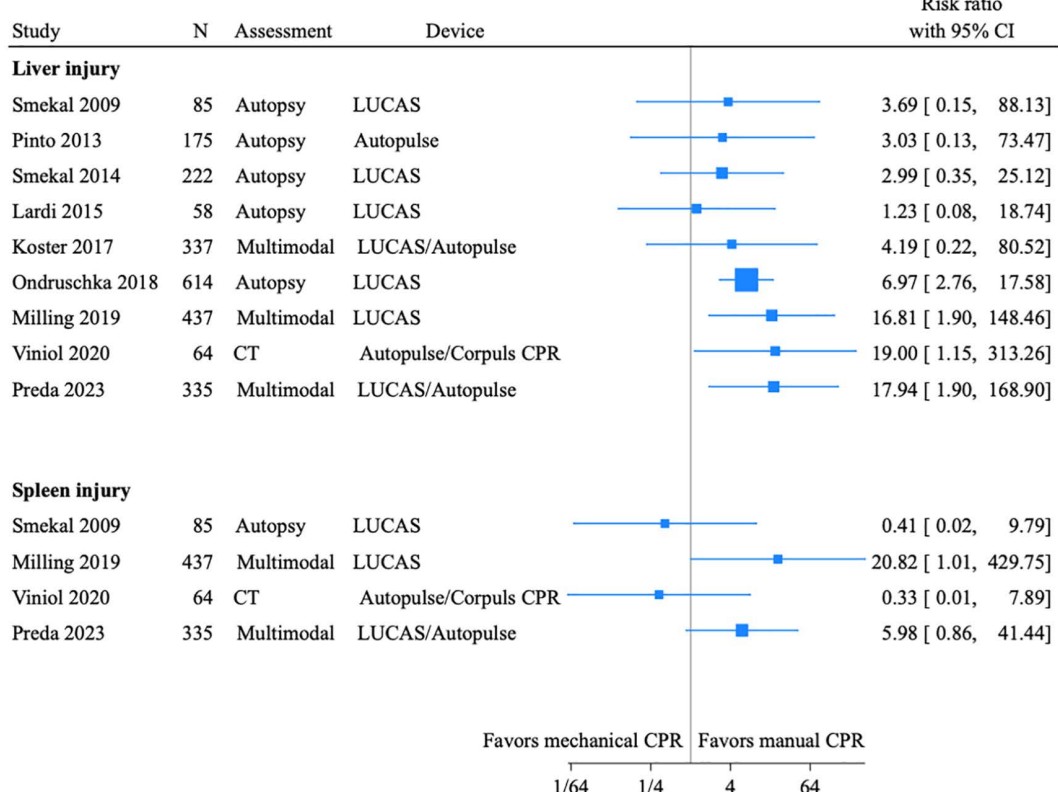

**Fig 3. Forest plot of studies comparing mechanical and manual chest compressions.** Unadjusted risk ratios for organ injury are presented for the two most commonly injured organs, liver and spleen. N is the number of patients with cardiac arrest in each study. CI: Confidence interval. LUCAS: Lund University Cardiopulmonary Assist System. CT: Computed tomography. CPR: Cardiopulmonary resuscitation.

## Outcome

Outcomes reported were mortality in ICU, in-hospital, at day 6, 28, 30 and 6 months, and neurological outcome at hospital discharge, day 28, 3 and 6 months. Poor neurological outcome was defined as Cerebral Performance Category (CPC) 3–5 in all studies. Few studies reported comparable outcomes for patients with and without CPR-related traumatic injury. Risk ratios for mortality in patients with severe compared to less severe abdominal ischaemia, as dichotomised in the studies, are presented as a Forest plot in Fig 4. Outcomes for each group are otherwise noted whenever available.

## Abdominal injury

**Abdominal injuries in general.** Several studies examined abdominal injuries using varied diagnostic methods including imaging (CT and/or ultrasound), and post-mortem examination (including post-mortem CT). Across these studies, the incidence of abdominal injuries were low, regardless of assessment modality and patient population.

Four studies employed multimodal examinations [14–17]. In a French registry of 1310 patients, six cases of CPR-related liver injury were identified, of whom three survived. One case of gastric perforation was reported, resulting in death [14]. In another registry, there were numerically more abdominal hemorrhages among patients treated with thrombolysis than without, but no difference in survival [15]. In a Czech cohort, all patients who died prior to hospital admission

| Study | N | Organ | Assessment | Mortality | Risk ratio with 95% CI |
|---|---|---|---|---|---|
| Oh 2015 | 148 | Liver | Aminotransferase | In-hospital | 2.34 [ 1.64, 3.35] |
| Champigneulle 2016 | 632 | Liver | Aminotransferase | ICU | 1.58 [ 1.40, 1.78] |
| Iesu 2018 | 374 | Liver | Aminotransferase | ICU | 1.84 [ 1.55, 2.18] |
| Wurm 2018 | 1780 | Intestines | Multimodal | In-hospital | 1.04 [ 0.59, 1.84] |
| Roedl 2019 | 1068 | Liver | Aminotransferase | 28-day | 1.46 [ 1.27, 1.69] |
| Paul 2020 | 1343 | Intestines | Multimodal | ICU | 1.66 [ 1.55, 1.76] |
| Schriefl 2021 | 156 | Intestines | Clinical signs | 6-month | 1.60 [ 0.88, 2.92] |
| Grimaldi 2022 | 159 | Stomach | Endoscopy | ICU | 1.43 [ 1.13, 1.80] |
| Farbu 2023 | 50 | Intestines | IFABP | 6-day | 25.00 [1.56, 400.54] |
| Tam 2023 | 597 | Intestines | CT | In-hospital | 1.48 [ 1.36, 1.60] |
| Delignette 2024 | 418 | Liver | PT-ratio | 28-day | 1.28 [ 1.20, 1.36] |
| *ECPR-patients:* | | | | | |
| Pang 2017 | 79 | Liver | Aminotransferase | In-hospital | 1.36 [ 1.09, 1.71] |
| Lee 2020 | 365 | Liver | Aminotransferase | In-hospital | 1.32 [ 1.11, 1.55] |
| Renaudier 2020 | 39 | Intestines | Multimodal | ICU | 1.21 [ 0.89, 1.64] |

Better with ischaemia | Worse with ischaemia

**Fig 4. Forest plot of studies reporting mortality for patients with abdominal ischaemia.** Unadjusted risk ratios are presented for patients with severe ischaemia compared to less severe ischaemia, as dichotomised in the study. CI: Confidence interval, ICU: Intensive care unit, CT: Computed tomography, PT-ratio: Prothrombin time ratio, ECPR: Extracorporeal cardiopulmonary resuscitation.

underwent autopsy, while there was no systematic screening for injuries in the intensive care unit (ICU). Liver injury was detected in 2.5% (16/628) of autopsies, compared to 1.7% (4/231) of ICU admissions; splenic injury was found in 1.8% (11/628) and 0.9% (2/231), respectively [17]. In a study from 1982, gastrointestinal haemorrhage occurred in 25 of 63 patients [16].

Five studies reported findings from abdominal CT within 6 hours of admission [18–22]. In one study, 363 of 597 patients underwent CT. Bowel ischaemia was identified in 24 patients, along with pancreatitis (n = 9), splenic or hepatic laceration (n = 7), hemoperitoneum (n = 4), and pneumoperitoneum (n = 1). None of the patients with bowel ischaemia survived. Outcomes related to abdominal injury were not consistently reported in other studies [18,20–22]. Less frequent findings included paralytic small bowel ileus, duodenal perforation, unspecific mesenteric lymphadenitis, ascites, acute cholecystitis, mesenteric vascular occlusion and ruptured abdominal aortic aneurism [19,20]. Among 104 patients, cardiac arrest was attributed to a perforated viscus in two cases and mesenteric ischemia in one [21]. In a study of Focused Assessment With Sonography in Trauma (FAST), five of 21 patients had free abdominal fluid, and none survived [23].

Two studies assessed CT findings not restricted to the early phase [24,25]. Pneumoperitoneum was observed in 1.7% and 4.5% of IHCA and OHCA cases, and haemoperitoneum in 3.4% and 0%, respectively [24]. The second study assessed the relationship between CPR compression depth and abdominal complications but did not report outcome data [25]. Two additional studies reported no association between thoracic anatomy and injury, and no injuries among 23 patients, respectively [26,27].

Among studies reporting solely post-mortem data, the incidence of liver injury ranged from 0.7% to 15%, and splenic injury from 0% to 3.8%. Most injuries were superficial (e.g., subcapsular tears or hematomas), although three were deemed serious and one life-threatening [28–33]. In one autopsy series from 1986, ischaemic bowel was considered a major diagnosis in nine of 130 patients, no other abdominal complications were noted [34].

**Mechanical compared to manual chest compressions.** In studies comparing mechanical and manual chest compressions, the reported incidence of liver injury ranged from 1.1% to 29.0% for mechanical, and 0% to 3.1% for manual compressions [35,37–42]. The incidence of spleen injury ranged from 0% to 2.4%, and 0% to 3.1%, respectively [35–37]. Unadjusted risk ratios for liver and spleen injury are presented in Fig 4. Due to a high level of heterogeneity, a quantitative synthesis was not feasible. No studies reported outcomes separately for patients with abdominal injuries.

In the study reporting an incidence of 29.0% for liver injury in the mechanical group, the resuscitation time was nearly twice the manual group, and most of the 32 patients were treated with a load-distributing band system [37]. In another comparison of load-distributing band system with manual compressions, there were haemoperitoneum in 17% (40/241) and 5% (4/82; p = 0.004), and pneumoperitoneum in 3% (7/241) and 0% (0/82; p = 0.13), respectively [43].

A Danish study of 1440 OHCA patients evaluated a piston-type compression device (Lund University Cardiopulmonary Assist System, LUCAS). Of the 207 patients treated with LUCAS, 84 sustained injuries, including four liver contusions, two kidney contusions, two spleen ruptures, one pancreas contusion, one gastric rupture, and one haemoperitoneum. Among patients treated with only manual compressions, a single liver contusion was reported. After adjusting for CPR duration, the overall incidence of injuries, including non-abdominal injuries, did not differ significantly between groups [36]. In a RCT comparing LUCAS and manual compressions, liver parenchymal injury occurred in 3.6% (5/139) and 1.2% (1/83), respectively, in a subset of patients with similar CPR duration [44]. In another two studies, two of thirteen patients treated with LUCAS during angiography developed liver haematomas, whereas no liver lacerations were found in mechanically resuscitated patients in a 1978 series [45,46].

**Organ specific studies.** Liver injury was assessed in six cohort studies (Table 1) [2,3,47–50]. Major CPR-related liver injury was found in 0.6% (15/2558), and nearly half (7/15) was considered life-threatening [47]. Hypoxic hepatitis was found in 7.2% to 20.5% of patients, and was consistently associated with mortality [2,48–50]. Acute liver failure was observed in 16.0% to 55.6% of patients and was associated with mortality in one study, while the other did not report mortality outcomes separately [3,48]. Seventy-five case reports described liver injury. Most cases occured after manual CPR, though some were after mechanical compressions, peri-arrest thrombolysis, and ECPR. Ultrasonography frequently revealed abdominal free fluid, and some patients had profound hemorrhagic shock. Management strategies included conservative treatment, angioembolisation, and surgical intervention via laparotomy (S3 Appendix).

Gastric mucosal lesions were assessed in three studies using endoscopy and autopsy (Table 1) [4,51,52]. Endoscopic findings revealed mucosal lesions to be common, however, only the most severe lesions were associated with increased mortality in the study by Grimaldi and collegues [4,51]. A total of 72 case reports described gastric rupture, typically involving the lesser curvature and resulting in pneumoperitoneum. Four cases presented with tension pneumoperitoneum characterized by hypotension and tachycardia. Several cases of gastric rupture occurred following brief periods of with mouth-to-mouth ventilation by lay responders. While some patients were managed conservatively, the majority required surgical intervention. 27 of the 72 patients died (S3 Appendix).

Intestinal ischaemia was assessed using clinical signs, biomarkers and multimodal diagnostic approaches, including CT, endoscopy, and surgical exploration (Table 1) [5–7,53–56]. Early diarrhoea was reported in 9.6% of patients. Non-occlusive mesenteric ischaemia (NOMI) occurred in 0.7% to 2.5% of cases. Elevated levels of intestinal fatty acid binding protein (IFABP), a biomarker of enterocyte injury, were observed in 82.0% to 100% of patients [5,7,54–56]. Mortality outcomes varied. One study reported no significant difference in mortality between patients with and without NOMI, whereas another reported a mortality rate of 96% among patients with suspected or confirmed NOMI [54,55]. Mortality was also significantly higher in patients with diarrhoea and elevated IFABP levels [5–7]. Thirteen case reports described intestinal ischemia, with additional reports of intestinal bleeding and transection. Most patients underwent surgical management with bowel resection, except in cases where further intervention was considered futile (S3 Appendix).

No cohort studies were identified that assessed splenic or pancreatic injury following CPR. However, 25 case reports described splenic rupture and six pancreatic injury. Most cases occured in conjunction with injuries to the liver or stomach.

**Table 1. Organ specific studies. Incidence and outcome.**

| Organ | Study (Author year) | N | Type of injury (Assessment modality) | Incidence % (proportion) | Organ specific outcome (with injury vs without) |
|---|---|---|---|---|---|
| *Liver* | Meron 2007 | 2558 | Major liver injury (Hospital record, surgery, autopsy)* | 0.6% (15/2558) | Considered life-threatening in 7/15 |
| | Oh 2015 | 148 | Hypoxic hepatitis (aminotransferase)† | 13.5% (20/148) | *Mortality, in-hospital*: 75.0% (15/20) vs 32.0% (41/128), p<0.001 *Poor neurological outcome, at discharge*: 100.0% (20/20) vs 58.6% (75/128) |
| | Champigneulle 2016 | 632 | Hypoxic hepatitis (aminotransferase)† | 11.4% (72/632) | *Mortality, ICU*: 86.1% (62/72) vs 54.5% (305/560), p<0.01 |
| | Iesu 2018 | 374 | Hypoxic hepatitis (aminotransferase)† | 7.2% (27/374) | *Mortality, ICU*: 88.9% (24/27) vs 51.6% (95/184) with liver failure but without hypoxic hepatitis, vs 44.8% (73/163) without hepatitis and failure, p=0.03 |
| | | | Acute liver failure (Prothrombin time ratio, bilirubin) | 55.6% (208/374) | *Poor neurological outcome, 3-months*: 92.6% (25/27) with hypoxic hepatitis vs 60.0% (201/347) without |
| | Roedl 2019 | 1068 | Hypoxic hepatitis (aminotransferase)† | 20.5% (219/1068) | *Mortality, day 28*: 57.1% (125/219) vs 39.0% (331/849), p<0.001 *Poor neurological outcome, day 28*: 64.8% (142/219) vs 52.1% (442/849), p<0.001 |
| | Delignette 2024 | 418 | Hypoxic hepatitis (aminotransferase)† | 14.6% (61/418) | *Mortality, day 28*: 95.1% (58/61) vs 78.2% (279/357), p<0.01 |
| | | | Acute liver failure (Prothrombin time ratio) | 16.0% (67/418) | 98.5% (66/67) vs 77.2% (271/351), p<0.001 |
| | Case reports | 75 | Liver injury (CT, surgery, autopsy) | | 30 patients died. 28/75 were treated with mechanical compression devices. |
| *Stomach* | McDonnell 1984 | 250 | Mucosal tears (Autopsy) | 2.0% (5/250) | NA |
| | L´Her 2005 | 130 | Gut mucosal lesions (endoscopy)‡ | All 36 endoscopies showed lesions. 60.0% (78/130) had signs of gut dysfunction | There was no correlation between gut lesion severity and outcome (not defined) |
| | Grimaldi 2022 | 214 | Gut mucosal lesions (endoscopy)§ | 56.5% (121/214) | *Mortality, ICU*: 70.9% (39/55) with severe vs 50.0% (33/66) with moderate vs 49.5% (46/93) without lesions, p=0.02 *Poor neurological outcome, at discharge*: 72.7% (40/55) with severe vs 56.1% (37/66) with moderate vs 59.1% (55/93) without lesions, p=0.14 |
| | Case reports | 72 | Stomach rupture (CT, x-ray, surgery, autopsy) | | 27 patients died. 2/72 were treated with mechanical compression devices. |
| *Intestines* | Grimaldi 2013 | 21 | Intestinal injury (plasma citrulline, urinary IFABP; pathologic levels) | *Citrulline*: 90.5% (19/21) *IFABP*: 100.0% (21/21) | NA |
| | Piton 2015 | 69 | Intestinal injury (plasma citrulline, plasma IFABP; pathologic levels)¶ | *Citrulline*: 82.4% (42/51) *IFABP*: 82.4% (42/51) | *Poor neurological outcome, 28-day*: Citrulline, below vs above: OR 9.8 (95% CI, 2.0 to 48.2), p=0.005 IFABP, above vs below: OR 4.2 (95% CI, 1.04 to 17.0), p=0.04 |
| | Wurm 2018 | 1780 | NOMI (NA)‖ | 0.7% (12/1780) | *Mortality, in-hospital*: 50.0% (6/12) vs 52.0% (919/1768) |

*(Continued)*

**Table 1.** (Continued)

| Organ | Study (Author year) | N | Type of injury (Assessment modality) | Incidence % (proportion) | Organ specific outcome (with injury vs without) |
|---|---|---|---|---|---|
| | Paul 2020 | 1343 | NOMI (CT, endoscopy, and/or surgery)** | 2.5% (33/1343) confirmed NOMI | *Mortality, ICU:* 96.3% (79/82) with suspected or confirmed NOMI vs 58.2% (734/1261) without |
| | Krychtiuk 2020 | 53 | Intestinal injury (plasma IFABP)†† | NA | *Mortality, 6-months*: aHR 2.72 (95% CI, 1.00 to 7.40), p=0.049 *Poor neurological outcome, 6-month:* No correlation (R=0.18, p=0.21) |
| | Schriefl 2021 | 156 | Early diarrhea (clinical signs)‡‡ | 9.6% (15/156) | *Mortality, 6-months*: 46.7% (7/15) vs 29.1% (41/141), p=0.24 *Poor neurological outcome, 6-month*: 66.7% (10/15) vs 36.9% (52/141), p=0.049 |
| | Farbu 2023 | 50 | Intestinal injury (plasma IFABP; pathologic levels)§§ | 100.0% (50/50) | *Mortality, day 30:* Per SD increase in IFABP: aOR 16.9 (95% CI, 1.1 to 261.3), p=0.04 *Poor neurological outcome, at discharge:* Above vs below median IFABP: 68.0% (17/25) vs 8.0% (2/25), p<0.001 |
| | Case reports | 15 | Intestinal injury (CT, endoscopy, surgery, autopsy) | | 10 patients died. 6/15 underwent surgery with resection of ischaemic or injured bowel segment. |
| *Spleen* | Case reports | 28 | Rupture of spleen (CT, surgery, autopsy) | | 15 died. 8 were treated with mechanical and 1 with active compression-decompression CPR, 6 with systemic and 1 with local thrombolysis. 12 were treated with splenectomy, 4 with embolisation, and 5 were dead before an intervention. |
| *Pancreas* | Case reports | 6 | Traumatic and ischaemic injury (Biomarker, CT, laparotomy, autopsy) | | 3 died. 1 patient was treated with mechanical CPR, none with thrombolysis. They were managed conservatively, by drainage or surgery. |

Table 1. Incidence of injury and outome in studies of specific abdominal organs. Poor neurological outcome was defined as Cerebral performance category (CPC) 3-5 in all studies. CT: Computed tomography, OR: odds ratio, CI: Confidence interval, ICU: Intensive care unit, IFABP: Intestinal fatty acid binding protein, aOR: adjusted OR, aHR: adjusted Hazard ratio, NOMI: Non-occlusive mesenteric ischaemia, NA: Not available/applicable, SD: Standard deviation, CPR: Cardio-pulmonary resuscitation.

*Major liver injury: anatomically defined as rupture/laceration of the liver capsule and/or subcapsular or intraparenchymal haematoma/haemorrhage.

†Hypoxic hepatitis: elevation of aminotransferase levels to at least 20-fold the upper limit of normal.

‡Inclusion was survival past 48 hours. Gut dysfunction was defined as: hiccough, emesis, enteral nutrition intolerance, pseudoobstruction, diarrhoea, hematemesis, melaena, proctorrhagia and/or acute haemoglobin lowering without any other identified cause.

§Gut mucosal lesion was defined as severe if there was ulceration or necrosis, and moderate if there was mucosal edema or erythema.

¶Low citrulline levels and high IFABP levels are pathological. Receiver operating characteristic curve analysis was applied to find the optimal cutoff value for citrulline and I-FABP concentrations for the prediction of 28-day CPC score. Presented ORs are below vs above citrulline concentration of 13.1 mmol/L, and above vs below plasma I-FABP of 260 pg/mL at ICU admission.

‖Suspected NOMI: clinical signs and symptoms or radiology findings and no sign of arterial occlusion.

Confirmed NOMI: two physicians reviewed the cases to confirm or exclude the presence of NOMI.

**Suspected NOMI: digestive symptoms and systemic signs such as shock or a lactic acidosis.

Confirmed NOMI: CT, endoscopy, and/or surgery.

††Above vs below plasma levels of IFABP ≥1500 pg/mL, determined by the Youden index. It was adjusted for time to ROSC and dose of adrenaline in the multivariable analysis

‡‡Early diarrhea: at least 2 liquid stools within the first 12 hours after ROSC.

§§It was adjusted for time to ROSC and lactate at admission in the multivariable analysis.

Splenic bleeding was typically managed with splenectomy, and thirteen of the 25 patients died. Pancreatic injuries were reported as either ischaemic and traumatic in origin. One case occurred following mechanical CPR. Management strategies included conservative treatment and drainage. Three of the six patients with pancreatic injury died (S3 Appendix).

**Abdominal injuries in ECPR-patients.** Eight studies reported abdominal injuries in patients undergoing ECPR (Table 2) [57–64]. The overall incidence of traumatic injury was 10.7% (11/103), and intra-abdominal haemorrhage occurred in 3.4% (3/87), with no significant difference in survival between patients with and without these complications [57,58]. Among 37 reported deaths, intra-abdominal bleeding was identified as the cause in two cases, and bowel infarction in one [59]. Intestinal ischaemia was reported in two studies, with an incidence of 12.8% (5/39) overall, and 5.6% (5/90) among patients receiving enteral nutrition [60,62]. In one study, all patients with intestinal ischaemia died, contributing to an overall ICU mortality rate of 56% (22/39) [60]. Hypoxic hepatitis occurred in 24.7% to 27.8% of patients and was identified as an independent predictor of mortality in both studies reporting this outcome [61,63].

**Abdominal injuries in children.** Abdominal injuries in children were evaluated in three post-mortem studies (Table 3). Among 211 children with a mean age of 19 months, one case of gastric perforation was identified, but no abdominal injuries were reported in the other two studies [65–67]. In a retrospective review of neonate medical records, the incidence of intestinal perforation was significantly higher in infants weighting more than 1000g who received CPR compared to those who did not (3.6% [7/194] vs 1.3% [73/5564]; p<0.01). No significant difference was observed in infants weighing less than 1000g (p=0.84). Clinical outcomes were not reported [68]. In two cohorts of children treated with ECPR, a hepatic dysfunction score was worse among non-survivors, and all patients with hepatic failure died [69,70].

**Table 2. Studies of ECPR with abdominal injuries reported.**

| Study (Author year) | N* | Type of injury (Assessment modality) | Incidence % (proportion) | Outcome (with injury vs without) |
|---|---|---|---|---|
| Pang 2017 | 79 | Hypoxic hepatitis (aminotransferase)† | 27.8% (22/79) | *Mortality, in-hospital:* 90.9% (20/22) vs 66.7% (38/57), p=0.03 |
| Maruhashi 2018 | 84 | Intraperitoneal hemorrhage (NA) | 3.6% (3/84) | NA |
| Zotzmann 2020 | 103 | Blunt abdominal trauma (whole-body CT within 24 hours) | 10.7% (11/103), mostly liver and spleen hematomas | *Mortality, in-hospital:* 81.8% (9/11) vs 87.0% (80/92), p=0.64 |
| Renaudier 2020 | 39 | Acute mesenteric ischaemia (Endoscopy, CT, or perioperative findings) | 12.8% (5/39) | *Mortality, ICU:* 100.0% (5/5) with acute mesenteric ischaemia |
| Lee 2020 | 365 | Hypoxic hepatitis (Aminotransferase)† | 24.7% (90/365) | *Mortality, in-hospital:* 72.2% (65/90) vs 54.9% (151/275), p=0.004 *Poor neurological outcome, at discharge:* 77.8% vs 63.6% |
| Gutierrez 2021 | 142 | Intestinal ischaemia, gastrointestinal bleeding (Medical records) | Among patients who received enteral feeding (n=90): 5.6% (5/90) intestinal ischaemia, 4.4% (4/90) gastrointestinal bleeding | NA |
| Bartos 2018 (nested within Gutierrez 2021) | (100) | Bleeding and bowel ischaemia (Endoscopy, CT) | 1.0% (1/100) had upper gastrointestinal bleeding | The patient with gastrointestinal bleeding survived. 37/100 patients died overall, of which 2/37 died of intra-abdominal bleeding and 1/37 of bowel infarction |

Table 2. Incidence of abdominal injury and outome in studies of ECPR patients. ECPR: Extracorporeal cardiopulmonary resuscitation, NA: Not available, CT: Computer tomography, CI: Confidence interval, OR: Odds ratio.

*N is the number of patients with refractory cardiac arrest.

†Hypoxic hepatitis was defined as elevation of aminotransferase levels to at least 20-fold the upper limit of normal.

**Table 3. Paediatric studies with abdominal injuries reported.**

| Study (Author year) | N | Population | Type of injury (Assessment modality) | Incidence % (proportion) | Outcome (with injury vs without) |
|---|---|---|---|---|---|
| Bush 1996 | 211 | Mean age 19.0 months | Gastric perforation (Autopsy) | 7/211 had injuries that were considered medically significant, of which one had gastric perforation. | NA |
| Price 2000 | 324 | <10 years | Abdominal injury (Autopsy) | 0% (0/324) | NA |
| Matshes 2010 | 546 | <18 years | Abdominal injury (Autopsy) | 0% (0/546) | NA |
| Soraisham 2014 | 8033 | Neonates, <33 weeks | Intestinal perforation (Medical records) | *Infants <1000g:* With CPR 4.0% (9/225). Without CPR: 5.4% (110/2050), (p=0.84) *Infants >1000g:* With CPR 3.6% (7/194) Without CPR 1.3% (73/5564), p<0.01 | NA |
| Kramer 2020 | 72 | ECPR, <18 years | Hepatic dysfunction (Hepatic dysfunction score at 48 hours) | All: 2 (IQR, 1–4) | Hepatic dysfunction score in survivors was 1 (IQR, 1–2) compared to 2 (IQR, 1–4) in non-survivors, p=0.025 |
| Ozturk 2022 | 26 | ECPR, <18 years* | Hepatic failure (Bilirubin)† | 19.2% (5/26) | *Mortality, in-hospital:* 100% (5/5) vs 57.1% (12/21), p=0.13 |

Table 3. Incidence of abdominal injury and outome in studies of paediatric patients. NA: Not available, CPR: cardio-pulmonary resuscitation, ECPR: Extracorporeal CPR, IQR: Interquartile range.

\* The cohort included 2 adults.

† Hepatic failure was defined as total bilirubin >10 mg/dL or conjugated bilirubin >3 mg/dL.

**Miscellaneous.** In a retrospective study of 9408 peripartum women, eleven experienced cardiac arrest. Among the three women with liver injury, two died [71]. In a study of liver transplantation, transaminase levels were elevated in donors with a history of cardiac arrest, however, neither the occurrence of cardiac arrest nor the degree of transaminase elevation correlated with graft failure [72]. In three studies evaluating interposed abdominal compressions during CPR, no abdominal injuries were reported [73–75].

## Discussion

In this systematic review, we identified 208 reports of abdominal injury following cardiac arrest, including 140 case reports. Many cohort studies were predominantly single-centre and focused on specific sub-populations, contributing to substantial heterogeneity. Overall, the incidence of CPR-related traumatic abdominal injuries was low, with liver and spleen being the most frequently affected organs. Although these injuries were potentially life-threatening, most were managed successfully through conservative treatment, surgery or angioembolisation, and did not appear to influence overall outcomes. Ischaemic complications demonstrated a broader range of incidence, largely dependent on the diagnostic modality employed. NOMI was reported in less than 3% of patients, while ischaemic liver injury occurred in 7% to 28%. Biomarkers indicative of intestinal injury, such as IFABP, were elevated in 82–100% of patients. Ischaemic abdominal injuries were consistently associated with increased mortality, regardless of assessment method.

The pattern of traumatic injuries observed resembled that of blunt abdominal trauma in general, with the liver and spleen most commonly affected, followed by hollow viscus and pancreatic injuries [77]. Unlike high-energy blunt trauma, CPR represents repetitive low-energy trauma, which may explain the low incidence of abdominal injuries [77,78]. An

exception was noted in peripartum women, with cardiac arrest, where nearly 30% of those who experienced cardiac arrest sustained liver injury [71]. The primary impact of chest compressions is the sternum, and accordingly the incidences of abdominal injuries were lower than that reported for thoracic injuries [79]. While traumatic abdominal injuries were not associated with population-level outcomes, several cases were life-threatening, underscoring the importance of clinical vigilance in patients who deteriorate after initial resuscitation. However, the rarity of severe injuries does not support routine abdominal imaging in all patients following CPR.

Several studies reported a higher incidence of abdominal organ injury with mechanical compared to manual chest compressions. In one study involving a load-distributing band system, liver injury occurred in 29% of patients [37]. Although a trial of this device was previously terminated early due to safety concerns, abdominal complications were not specifically reported [80]. Thoracic injuries have also been more frequently reported with mechanical compared to manual compressions [79]. However, patients receiving mechanical compressions often have prolonged resuscitation times, which may confound these findings. Notably, three RCT´s comparing mechanical and manual chest compressions reported no significant difference in survival [81–83]. Only the LINC-trial reported abdominal adverse events separately [44,81–83]. Despite the lack of systematic assessment, current evidence does not suggest that mechanical compressions significantly impact outcomes via increased abdominal injury.

Ischaemic abdominal injuries showed highly variable incidence rates, depending on diagnostic method. This variability may reflect both the subtlety of ischaemic injury and the inherent difficulty in diagnosis [84–86]. Ischaemia is common in other organs following cardiac arrest, acute kidney injury occurs in 50% of patients, and hypoxic-ischaemic brain injury remains the leading cause of death [1,87]. Given that all patients likely experience some degree of ischaemia during cardiac arrest, even with high-quality CPR, ischaemic injury may represent a continuum rather than a binary event [1]. Biomarkers like IFABP may offer greater sensitivity, although their clinical utility remains uncertain [2,7]. For instance, IFABP was not predictive of outcomes in patients with refractory cardiac arrest in a recent study, and was inconsistently associated with clinical manifestations of NOMI [88]. Future research should aim to enhance diagnostic strategies for abdominal ischaemia and explore the timing and mechanisms by which abdominal injury contributes to organ dysfunction and mortality [89].

Determining the true incidence of abdominal injury is challenging. In addition to diagnostic limitations, assessing only sub-populations might lead to both under- and overestimation [85]. For example, injuries may be missed in patients who do not undergo imaging, while autopsy-based studies might overestimate the incidence of severe injuries in non-survivors [4,20,90,91].

Most studies reported an association between abdominal ischaemia and increased mortality. We presented risk ratios for mortality in a Forest plot to illustrate this relationship. However, interpretation requires caution. First, defining a threshold for ischaemic injury is inherently problematic given its continuum. Second, confounding factors such as time to return of spontaneous circulation (ROSC) were not adjusted for, raising the possibility that abdominal injury may simply reflect whole-body ischaemia. Accordingly, severe intestinal ischaemia requiring surgical resection was uncommon [54,55]. However, a recent study reported that clinical signs of NOMI was associated with poor neurological outcomes in an adjusted analysis [92].

Ischaemic abdominal injury may contribute to poor outcomes by adding to the total ischaemic burden and triggering systemic inflammation [1]. Although several trials explored anti-inflammatory therapies after cardiac arrest, none have demonstrated clear benefits on patient-centered outcomes [93–95]. Our previous work suggest that the inflammatory consequences of intestinal injury may be limited [96]. Nonetheless, the clinical impact of abdominal ischaemia after cardiac arrest remains uncertain. Emerging interventions such as resuscitative endovascular balloon occlusion of the aorta (REBOA), which may aggravate abdominal ischaemia, should be implemented cautiously [97,98].

Our review has several limitations. Despite a comprehensive search strategy, some studies were identified through hand-searching, and grey literature was not included. Most included studies were of low to moderate quality with variability

in patient populations, exposures, and outcomes. Many lacked clear inclusion criteria or standardized definitions, complicating synthesis. We categorized injuries as traumatic or ischaemic for the ease of interpretation, although this dichotomy it is not always clinically or pathophysiologically distinct.

## Conclusion

In this comprehensive review of abdominal injuries following cardiac arrest, CPR-related traumatic injuries were uncommon, but should be considered in patients with unexplained clinical deterioration. Ischaemic injury incidence ranged from 0.7% to 100%, and was consistently associated with increased mortality. Whether abdominal ischaemia independently contributes to poor outcomes remains unresolved and warrants further investigation.

## Supporting information

**S1 Appendix. PRISMA Checklist.**
(DOCX)

**S2 Appendix. The complete search strategy.**
(DOCX)

**S3 Appendix. References to all case reports and case series.**
(DOCX)

**S4 Appendix. Quality appraisal of included studies.**
(DOCX)

**S5 Appendix. Protocol.**
(DOCX)

**S6 Table. All included studies except case reports.**
(DOCX)

**S7 File. Included studies.** Excel spreadsheat.
(XLSX)

**S8 File. Excluded studies with reasons.** Excel spreadsheat.
(XLSX)

**S9 Fig. Data supporting figure in excel spreadsheat.**
(XLSX)

**S10 Fig. Data supporting figure in excel spreadsheat.**
(XLSX)

**S11 Fig. Data supporting figure in excel spreadsheat.**
(XLSX)

## Acknowledgments

We would like to thank Ingrid Ingeborg Riphagen, Katrine Aronsen and the other staff at the Medicine and Health Library, Norwegian University of Science and Technology, for their assistance and cooperation with this systematic review.

## Author contributions

**Conceptualization:** Bjørn Hoftun Farbu, Jostein Hagemo, Marius Rehn.

**Data curation:** Bjørn Hoftun Farbu, Marius Rehn.

**Formal analysis:** Bjørn Hoftun Farbu, Marius Rehn.

**Investigation:** Bjørn Hoftun Farbu, Jostein Hagemo, Marius Rehn.

**Methodology:** Bjørn Hoftun Farbu, Jostein Hagemo, Marius Rehn.

**Project administration:** Bjørn Hoftun Farbu.

**Supervision:** Marius Rehn.

**Visualization:** Bjørn Hoftun Farbu.

**Writing – original draft:** Bjørn Hoftun Farbu.

**Writing – review & editing:** Bjørn Hoftun Farbu, Jostein Hagemo, Marius Rehn.

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
