## [Decision Letter · Decision Letter 0]

PONE-D-25-08582Abdominal organ injury in cardiac arrest: Systematic literature reviewPLOS ONE

Dear Dr. Hoftun Farbu,

Thank you for submitting your manuscript to PLOS ONE. After careful consideration, we feel that it has merit but does not fully meet PLOS ONE’s publication criteria as it currently stands. Therefore, we invite you to submit a revised version of the manuscript that addresses the points raised during the review process.

We look forward to receiving your revised manuscript.

Kind regards,

Nik Hisamuddin Nik Ab. Rahman

Academic Editor

PLOS ONE

Journal Requirements:

3. We note that your Data Availability Statement is currently as follows: All relevant data are within the manuscript and in Supporting Information files.

4. As required by our policy on Data Availability, please ensure your manuscript or supplementary information includes the following:

Reviewers' comments:

Reviewer's Responses to Questions

**Comments to the Author**

1. Is the manuscript technically sound, and do the data support the conclusions?

Reviewer #1: Yes

Reviewer #2: Yes

2. Has the statistical analysis been performed appropriately and rigorously? 

Reviewer #1: N/A

Reviewer #2: Yes

3. Have the authors made all data underlying the findings in their manuscript fully available?

Reviewer #1: Yes

Reviewer #2: Yes

4. Is the manuscript presented in an intelligible fashion and written in standard English?

Reviewer #1: No

Reviewer #2: Yes

5. Review Comments to the Author

Reviewer #1: Dear Author

May your day be filled with sunshines and smiles. I read your paper. It discusses an interesting topic. The complications of abdominal organ injury following resuscitation are an important and interesting topic. It requires the following minor revisions to prepare it for further consideration. My opinion is "Minor Revision".

1. English needs revision.

2. All keywords should be selected from Mesh Databases.

Best regards

Reviewer #2: I am grateful for the opportunity to review this manuscript. The following is a detailed evaluation based on the PRISMA 2020 checklist and the requirements of PLOS ONE, followed by a decision regarding its suitability for publication. As a scientific reviewer, the sections of the manuscript requiring revision, based on the evaluation conducted, are outlined below in order:

• Only one reviewer conducted data extraction and quality assessment, potentially increasing the risk of bias. It is recommended that data extraction and quality assessment be performed independently by at least two reviewers, with discrepancies resolved through discussion or by a third reviewer. If this is not feasible, the limitations section should explicitly acknowledge that this approach may have introduced bias.

• Grey literature, such as unpublished reports or theses, was not searched, which may contribute to publication bias. The methods or limitations section should clearly state that grey literature was not reviewed, and note that this may have impacted the comprehensiveness of the review.

• High data heterogeneity prevented quantitative synthesis, limiting the ability to provide precise estimates. The discussion section should elaborate on the reasons for heterogeneity and its implications for the results. Future research should also prioritize standardizing definitions and assessment methods.

• While the search strategy is included in the appendix, it lacks sufficient detail in the main text. A concise paragraph summarizing the search strategy, including key terms and databases searched, should be added to the methods section.

• Definitions of traumatic and ischemic injuries were, in some instances, determined by the authors, potentially introducing bias. The methods section should clarify how injuries were classified, especially when definitions were absent in the original studies. Consideration of sensitivity analysis with alternative definitions is also advised.

• The characteristics of the included studies are described broadly without sufficient specificity. The results section should contain a table summarizing key study characteristics (e.g., study design, population, assessment methods, and outcomes).

• The discussion of clinical implications for abdominal injuries is underdeveloped. The discussion section should address how the findings might inform clinical management, such as whether routine assessment of abdominal injuries is warranted.

• Several references are outdated and require replacement with more current studies. References should be updated to incorporate recent, relevant studies, and citation accuracy should be confirmed.

• The logical progression in some sections is inconsistent or unclear. The manuscript’s structure should be refined to ensure seamless transitions between sections, such as linking results directly to their interpretation in the discussion.

• The conclusion lacks precision in light of the findings and limitations. The conclusion should briefly restate the main findings and offer specific recommendations for future research, such as prospective studies with standardized definitions.

These revisions are critical to enhancing the manuscript’s quality and aligning it with scientific standards. Following these updates, the manuscript should undergo re-evaluation.

6. PLOS authors have the option to publish the peer review history of their article (what does this mean? ). If published, this will include your full peer review and any attached files.

**Do you want your identity to be public for this peer review?** For information about this choice, including consent withdrawal, please see our Privacy Policy .

Reviewer #1: No

Reviewer #2: **Yes: ** Afshin Khazaei

---

## [Author Response · Author response to Decision Letter 1]

4 Jul 2025

Thank you for considering our manuscript for publication. We appreciate the comments, and have addressed all the concerns raised by editor and reviewers. Please find our point by point responses in blue text below.

We have now changed the following to meet PLOS ONE´s style requirements:

We have changed the formatting of the headings.

We have removed the use of returns to align content in Table 1

We have removed text from the figure file names and supplementary appendix names.

We have now removed all funding-related text from the manuscript.

3. We note that your Data Availability Statement is currently as follows: All relevant data are within the manuscript and in Supporting Information files.

We provide the data for Fig 2-4 and all extracted data in excel spreadsheats in Supporting Information files.

4. As required by our policy on Data Availability, please ensure your manuscript or supplementary information includes the following:

Due to the large number of studies, we provide two tables. One numbered table for studies excluded after review in full-text, with reasons for exclusion. We also provide one numbered table with all included studies. The table contains all data extracted and include name of data extractor and date of data extraction, and confirmation that the study was eligible to be included in the review. In addition, data obtained from one study by e-mail from the corresponding author is noted in the file (Renaudier 2020).

The complete risk of bias and quality assessment for all included studies is provided in the original submission (S4 Appendix). We have added the following phrase under «Statistical methods» regarding missing data: «Given the absence of a quantitative synthesis, only the data at hand were reported, and no formal steps were taken to address missing data.” All data extracted from each study that would be needed to replicate the figures are also provided as excel spreadsheats, as noted above.

Comments to the Author

5. Review Comments to the Author

Reviewer #1: Dear Author

May your day be filled with sunshines and smiles. I read your paper. It discusses an interesting topic. The complications of abdominal organ injury following resuscitation are an important and interesting topic. It requires the following minor revisions to prepare it for further consideration. My opinion is "Minor Revision".

1. English needs revision.

2. All keywords should be selected from Mesh Databases.

Best regards

Thank you for your time spent reading our manuscript. We are delighted that you find the topic both important and interesting. We have revised the English througout the manuscript, as shown in the manuscript with tracked changes. All keywords is now selected from Mesh Databases.

Reviewer #2: I am grateful for the opportunity to review this manuscript. The following is a detailed evaluation based on the PRISMA 2020 checklist and the requirements of PLOS ONE, followed by a decision regarding its suitability for publication. As a scientific reviewer, the sections of the manuscript requiring revision, based on the evaluation conducted, are outlined below in order:

• Only one reviewer conducted data extraction and quality assessment, potentially increasing the risk of bias. It is recommended that data extraction and quality assessment be performed independently by at least two reviewers, with discrepancies resolved through discussion or by a third reviewer. If this is not feasible, the limitations section should explicitly acknowledge that this approach may have introduced bias.

Thank you for this important comment. We planned to conduct quality assessment by two independent reviewers. After reviewing the studies in full-text however, it was clear that a meta-analysis was not appropriate. Given the large number of studies included, we found that it was not feasible for two reviewers to extract data and perform quality assessment, and made an amendment to protocol as stated in the manuscript. We have included the following phrase in the limitation section: «Data extraction and quality appraisal were performed by a single reviewer, introducing potential bias.”

• Grey literature, such as unpublished reports or theses, was not searched, which may contribute to publication bias. The methods or limitations section should clearly state that grey literature was not reviewed, and note that this may have impacted the comprehensiveness of the review.

As the reviewer point out, not reporting grey literature may lead to publication bias. A comprehensive review was one of our objectives, but we found that including grey literature would not add substantially value to this review. We have stated in the Methods section: “Grey literature was not searched.” Further, we added the phrase “Grey literature was not searched, and this may have impacted the comprehensiveness of the review” to the limitations section.

• High data heterogeneity prevented quantitative synthesis, limiting the ability to provide precise estimates. The discussion section should elaborate on the reasons for heterogeneity and its implications for the results. Future research should also prioritize standardizing definitions and assessment methods.

Thank you for this comment. We have rewritten parts of the Discussion section:

“Many cohort studies were predominantly single-centre and focused on specific sub-populations, contributing to substantial heterogeneity.”

“Ischaemic abdominal injuries showed highly variable incidence rates, depending on diagnostic method. This variability may reflect both the subtlety of ischaemic injury and the inherent difficulty in diagnosis”.

“Biomarkers like IFABP may offer greater sensitivity, although their clinical utility remains uncertain [2, 7]. For instance, IFABP was not predictive of outcomes in patients with refractory cardiac arrest in a recent study, and was inconsistently associated with clinical manifestations of NOMI [88].”

“Determining the true incidence of abdominal injury is challenging. In addition to diagnostic limitations, assessing only sub-populations might lead to both under- and overestimation [84]. For example, injuries may be missed in patients who do not undergo imaging, while autopsy-based studies might overestimate the incidence of severe injuries in non-survivors [4, 20][89, 90].“

Finally, we have added a sentence on future research: “Future research should aim to enhance diagnostic strategies for abdominal ischaemia and explore the timing and mechanisms by which abdominal injury contributes to organ dysfunction and mortality”

• While the search strategy is included in the appendix, it lacks sufficient detail in the main text. A concise paragraph summarizing the search strategy, including key terms and databases searched, should be added to the methods section.

In accordance with the comment from the reviewer, we describe the search strategy in the Methods section: “We searched MEDLINE/PubMed, Embase, and The Cochrane Database of Systematic Reviews up to 12th September 2024, and conducted a citing reference search in Scopus up to 15th November 2024.”

We have also included a concise paragraph summarizing the search strategy and key terms:

“The search strategy consisted of both controlled terms (Medical Subject Headings [MeSH], Emtree) and free text words for two basic concepts: 1. Cardiac arrest and 2. Abdominal organ injury. The first set of entry terms described the patient group, and the second set described the exposure. We used Boolean operators for term combinations and included synonyms. Proximity operators and field specification were applied when available and suitable.”

• Definitions of traumatic and ischemic injuries were, in some instances, determined by the authors, potentially introducing bias. The methods section should clarify how injuries were classified, especially when definitions were absent in the original studies. Consideration of sensitivity analysis with alternative definitions is also advised.

The reviewer raises an important issue. Determining whether an injury is traumatic or ischemic is not always straightforward, and sometimes it might be a combination of both. However, to present the large number of heterogenous studies, we found that it would help the reader if we stratified the injuries based on aetiology, since the two types of injuries require different clinical approaches. Sensitivity analyses and different stratification methods were considered, but given that meta-analyses were found not to be appropriate however, we decided not to include this in the review.

The following paragraph under the heading Definitions describes how injuries were classified: “Injuries were referred to as traumatic or ischaemic as defined in the studies. If not defined, the following injuries were referred to as traumatic: hollow viscus perforation, pneumoperitoneum, hemoperitoneum, tear, laceration, contusion, rupture, hematoma, and hemorrhage. Rise in biomarkers, e.g. transaminases, and mucosal lesions were referred to as ischaemic injury.”

We have also addressed this in the limitations section: “We categorized injuries as traumatic or ischaemic for the ease of interpretation, although this dichotomy it is not always clinically or pathophysiologically distinct.”.

• The characteristics of the included studies are described broadly without sufficient specificity. The results section should contain a table summarizing key study characteristics (e.g., study design, population, assessment methods, and outcomes).

We have now provided a table with key study characteristics (e.g., study design, population, assessment methods, and outcomes) for all included studies except case reports. Since this table is considerably large, we suggest adding this table as supplementary information.

• The discussion of clinical implications for abdominal injuries is underdeveloped. The discussion section should address how the findings might inform clinical management, such as whether routine assessment of abdominal injuries is warranted.

Thank you for this important comment. The clinical implications is crucial to address, and we have revised the manuscript accordingly. In the Discussion section, we write: «Overall, the incidence of CPR-related traumatic abdominal injuries was low, with liver and spleen being the most frequently affected organs. Although these injuries were potentially life-threatening, most were managed successfully through conservative treatment, surgery or angioembolisation, and did not appear to influence overall outcomes.”

Further, we have added: “While traumatic abdominal injuries were not associated with population-level outcomes, several cases were life-threatening, underscoring the importance of clinical vigilance in patients who deteriorate after initial resuscitation. However, the rarity of severe injuries does not support routine abdominal imaging in all patients following CPR..”

Regarding the use of mechanical chest compression devices, we have noted: “Despite the lack of systematic assessment, current evidence does not suggest that mechanical compressions significantly impact outcomes via increased abdominal injury.”

Further we write that “(…) severe intestinal ischaemia requiring surgical resection was uncommon”. And “Emerging interventions such as resuscitative endovascular balloon occlusion of the aorta (REBOA), which may aggravate abdominal ischaemia, should be implemented cautiously”.

• Several references are outdated and require replacement with more current studies. References should be updated to incorporate recent, relevant studies, and citation accuracy should be confirmed.

In accordance with this comment, we have updated the references.

• The logical progression in some sections is inconsistent or unclear. The manuscript’s structure should be refined to ensure seamless transitions between sections, such as linking results directly to their interpretation in the discussion.

We have made a substantial revision of the manuscript to enhance clarity, and avoid inconsistencies. Several paragraphs have been changed, and along with improved language we hope that the readers now may follow the progression in a better way.

• The conclusion lacks precision in light of the findings and limitations. The conclusion should briefly restate the main findings and offer specific recommendations for future research, such as prospective studies with standardized definitions.

These revisions are critical to enhancing the manuscript’s quality and aligning it with scientific standards. Following these updates, the manuscript should undergo re-evaluation.

Thank you for this comment. We have now revised the conclusion, which include the main findings and specific recommondations for future research:

“In this comprehensive review of abdominal injuries following cardiac arrest, CPR-related traumatic injuries were uncommon, but should be considered in patients with unexplained clinical deterioration. Ischaemic injury incidence ranged from 0.7% to 100%, and was consistently associated with increased mortality. Whether abdominal ischaemia independently contributes to poor outcomes remains unresolved and warrants further investigation.”

---

## [Editor Report · Decision Letter 1]

Abdominal organ injury in cardiac arrest: Systematic literature review

PONE-D-25-08582R1

Dear Dr. Farbu,

We’re pleased to inform you that your manuscript has been judged scientifically suitable for publication and will be formally accepted for publication once it meets all outstanding technical requirements.

Kind regards,

Nik Hisamuddin Nik Ab. Rahman

Academic Editor

PLOS ONE
---

## [Editor Report · Acceptance letter]

PONE-D-25-08582R1

PLOS ONE

Dear Dr. Hoftun Farbu,

I'm pleased to inform you that your manuscript has been deemed suitable for publication in PLOS ONE. Congratulations! Your manuscript is now being handed over to our production team.

Kind regards,

on behalf of

Professor Dr Nik Hisamuddin Nik Ab. Rahman

Academic Editor

PLOS ONE